

# Origin identification of migratory pests (European Starling) using geochemical fingerprinting

Upama Khatri-Chhetri[1], John G. Woods[2], Ian R. Walker[2] and P. Jeff Curtis[3]

[1] Agriculture, Food and Nutritional Science, University of Alberta, Edmonton, Alberta, Canada
[2] Biology, University of British Columbia, Kelowna, British Columbia, Canada
[3] Earth, Environmental and Geographic Sciences, University of British Columbia, Kelowna, British Columbia, Canada

## ABSTRACT

The European Starling (Sturnidae: *Sturnus vulgaris* L.) is an invasive bird in North America where it is an agricultural pest. In British Columbia (Canada), the starling population increases in orchards and vineyards in autumn, where they consume and damage ripening fruits. Starlings also cause damage in dairy farms and feedlots by consuming and contaminating food and spreading diseases. Damage can be partly mitigated by the use of scare devices, which can disperse flocks until they become habituated. Large-scale trapping and euthanizing before starlings move to fields and farms could be a practical means of preventing damage, but requires knowledge of natal origin. Within a small (20,831 km$^2$), agriculturally significant portion of south-central British Columbia, the Okanagan-Similkameen region, we used 21 trace elements in bone tissue to discriminate the spatial distribution of juvenile starlings and to reveal the geographic origin of the problem birds in fall. Stepwise discriminant analysis of trace elements classified juveniles to their natal origin (minimum discrimination distance of 12 km) with 79% accuracy. In vineyards and orchards, the majority (55%) of problem birds derive from northern portions of the valley; and the remaining 45% of problem birds were a mixture of local and immigrant/unassigned birds. In contrast, problem birds in dairy farms and feedlots were largely immigrants/unassigned (89%) and 11% were local from northern region of the valley. Moreover, elemental signatures can separate starling populations in the Valley yielding a promising tool for identifying the geographic origin of these migratory birds.

## INTRODUCTION

European starling (*Sturnus vulgaris* L.) is one of the most successful non-native species in North America (*Vuilleumier, 2009*). A few hundred birds were introduced in New York City in 1890 (*David, 2010*; *Kerpez & Smith, 1990*; *Linz et al., 2007*). Surviving descendant birds spread rapidly throughout North America in the following years (*Cabe, 1993*; *Linz et al., 2007*) and have now grown to a population of many millions. The perception of starlings

Corresponding authors
Upama Khatri-Chhetri,
upama@ualberta.ca
P. Jeff Curtis, jeff.curtis@ubc.ca

changed along with population size from a beautiful, robin size 'Chunky' songbird into an aggressive and costly pest.

Starling is listed as one of the three worst invasive birds in the World Conservation Union List of '100 of the world's worst invasive alien species' (*Lowe et al., 2004*). According to the International Union for Conservation of Nature, its conservation status is considered to be of least concern on the basis of its probability of extinction. It is considered as an agricultural pest, especially in fruit crops such as cherries, berries, and grapes (*Pimentel, Zuniga & Morrison, 2005*; *BCGA, 2013*; *Conover, Berryman & Dolbeer, 2007*; *Virgo, 1971*).

The migration pattern of European starlings varies regionally and individually. In some areas, most of the breeding individuals are sedentary (*Kessel, 1953*; *Suthers, 1978*). Some individuals migrate in some years and not in others. Some juveniles migrate, but even their nest mates (siblings) might not (*Kessel, 1953*). Generally, spring migration occurs between mid-February to end of March and fall migration occurs from September to December (*Dolbeer, 1982*; *Kessel, 1953*). Starlings can cover up to 1,000–1,500 km if needed in search of food especially during winter (*Linz et al., 2007*).

The Okanagan-Similkameen region is a nationally significant agricultural region in Canada due to climatic and geological suitability. The valley is ideal for several tree fruits such as apples, peaches, and other soft fruits, like cherries, berries, and grapes, and for livestock operations including both dairy farms and beef cattle feedlots. Around 97.5% of British Columbia's vineyard and 98% of the province's apples and grapes are grown in the south-central Okanagan-Similkameen region region (*Neuhauser, 2013*; *Growers, 2015*). In the northern Okanagana region, livestock operations, such as dairy farms and feedlots, grains and forage crops predominate (*Growers, 2015*; *BCGA, 2013*). The cold short winters, warm summers, rural/urban interface, and availability of year-round food sources in the Okanagan-Similkameen region provide excellent starling habitat.

Large flocks of up to 1,000s starlings are known to destroy fruits in the vineyards, orchards and deprecate cattle feed at dairy farms (*Glahn, 1981*). Annually around $800 million USD worth of agricultural crops were destroyed by starlings in the United States, based on crop losses of $5/ha (*Pimentel, 2000*). In livestock facilities, starlings consume and contaminate around 15–20 tons of cattle feed per day (*Linz et al., 2007*). Annually, around $3 million USD per year is lost in vineyards and tree fruit farms in the Okanagan-Similkameen region alone (*BCGA, 2012*).

A Starling Control Program (SCP) was initiated in the Okanagan-Similkameen region of British Columbia in 2003 (*BCGA, 2010*) to control the starling population and the damage they cause. Agricultural industries, environmental agencies, and regional districts supported the program (*BCGA, 2010*). The SCP traps birds in various locations in the Okanagan-Similkameen region throughout the year. From 2003 to 2013 around 544,000 birds were trapped by the control program (*OSSCP, 2014*). Although the control program has been trying to reduce the starling population by aggressively trapping all year round, numbers appear to be increasing from 996 to 23,592 from 1974 to 2013 (*National Audubon Society, 2016*), especially in the fall season due to migration and natal dispersal. Starling numbers increase in vineyards and orchards in fall (*BCGA, 2013*), but dispersion and movement patterns of starlings has not been well understood in the Okanagan-Similkameen region. It

is necessary to find the origin of these migrant birds to enhance program effectiveness. Thus, understanding the natal origin and movement pattern of migratory pests, like starlings, is important for developing successful management (*Cabe, 1993*; *Neuhauser, 2013*). The main objective of this research is to identify the origin of the migratory starlings using a geochemical (trace element) fingerprinting approach in the Okanagan-Similkameen region so that starling might be managed at sources.

Biogeochemical markers such as stable isotopes (*Hobson, 1999*; *Hobson, 2005a*; *Hobson, 2005b*; *Szép et al., 2009*) and trace elements analyses (*Szép et al., 2003*; *Szép et al., 2009*) are potential methods to trace the origin and migration of such birds because the markers are indicators of the environment of origin. Starlings can disperse long distances and retain stable isotopes and trace element signatures in tissues reflecting natal food (*Hobson, 1999*), location, and habitat (*Szép et al., 2009*). These chemical signatures can be applied as tools to reconstruct animal movement pathways (*Ethier et al., 2013*).

We chose trace elements over light isotope ratios for this study for two reasons. First, isotope ratios in birds can be homogeneous for thousands of kilometres (*Inger & Bearhop, 2008*), an order of magnitude larger than the Okanagan-Similkimeen region. In contrast, the province of British Columbia and Okanagan-Similkameen region are geochemically diverse across spatial scales of tens of kilometres or less (*Okulitch, 2013*), making trace elements potentially much more sensitive for tracking regional starling movements (*Neuhauser, 2013*). Second, trace elements can provide a higher specificity than isotopes because of the larger number of potential variables (*Szép et al., 2009*). We hypothesized that the elemental composition of starling bones depends on geographic region. Because birds acquire trace element signatures in their tissues from their diet, and these signatures reflect the chemical composition of the area in which the tissue was generated, or keep the record of past feeding history as recorded in long turnover tissues (*Hobson, 1999*; *Szép et al., 2009*).

We used samples of starling leg bone tissue (tarsometatarsus) commonly known as tarsus or metatarsus to fingerprint natal origins of individual starlings. Feather tissue is more convenient to use but we chose bone because feathers moult seasonally shedding the evidence of natal origin, whereas the turnover rate of bone tissue is as much as 30 years (*Tieszen et al., 1983*; *Lanocha & Kalisinska, 2012*). Bone tissues are very useful for origin analysis as the mineral phase in the bone matrix grows rapidly when an individual is a juvenile and then turns over very slowly (*Neuhauser, 2013*). Most bone mass consists of type I collagen and apatite (a mineral composed mainly of calcium and phosphate) (*Boonen et al., 2010*). The bone mineralization is analogous to the geological mineral (*Boonen et al., 2010*) and is known as hydroxyapatite, $Ca_5 (PO_4)_3OH$ (*Boskey, 2007*; *Pasteris, Wopenka & Valsami-Jones, 2008*). Other cations and anions easily substitute into the mineral matrix of apatite. Thus, other elements, derived from the diet can be incorporated into bone to create a fingerprint, representing the food an individual has ingested (*Grynpas et al., 1993*). The decision of using bone is also supported by an earlier study of six European starling tissues (bone, liver, heart, muscle, brain, and feather) that identified bone as the tissue for statistically separating populations of starlings across much of its range in British Columbia (*Neuhauser, 2013*).

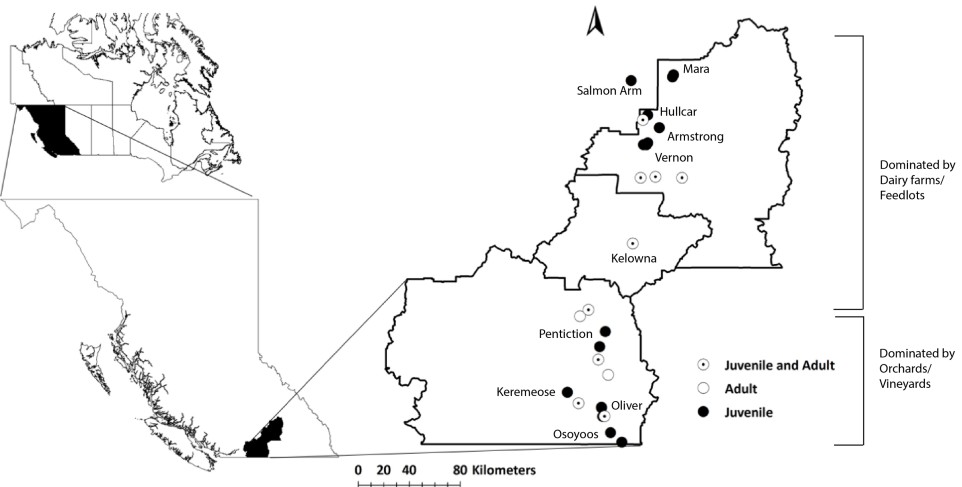

**Figure 1  Map of the study area (Okanagan-Similkameen region) with sampling sites in British Columbia (BC), Canada.** Filled black circles represent sampling sites for juvenile starlings, whereas unfilled black circles represent sampling sites for adult starlings. Black circles with black dots represent the sampling sites where both adults and juveniles were collected.

In this study, we evaluated the geochemical fingerprint of juvenile starlings throughout the Okanagan-Similkameen region, and compared these signatures to that of problem birds caught in the fall trapped in fruit crops in and dairy farms. By fingerprinting starling natal origin, appropriate management to reduce starling populations could be implemented in focused areas to reduce the damage created by starlings.

# MATERIAL AND METHODS

## Study Area

British Columbia (BC) is the western-most province of Canada and has high geological diversity in rock types (*Natural Resources Canada, 2012*) (Fig. 1). The Okanagan-Similkameen region in south-central British Columbia is about 200 km long and 20 km wide (Fig. 1). It lies in between the Columbia and Cascade Mountain ranges in the southern interior of BC. The valley bedrock is comprised of volcanic, sedimentary, metamorphic and intrusive rocks (*Okulitch, 2013*). Southern Okanagan-Similkameen region is heterogeneous, made up of volcanic, metamorphic, plutonic, and sedimentary rocks, and overlain by glacial sediments (*Okulitch, 2013*). Central Okanagan is made up lava, andesite with quartz-filled amygdales. The northern Okanagan is made up of Paleozoic phyllite, argillite, schist, amphibolitic, quartzite, carbonaceous, micaceous chlorite and other minor constituents (Table S1) (*Okulitch, 2013*).

Starling habitat is generally the lower elevation portion of a long valley, approximately between 300 and 700 m elevation. Above 700 m, the valley rises quickly to a plateau of >1,200 m (Fig. S1), greatly constraining distribution and likely movement and migration to a relatively small area (Fig. S1).

## Sampling

Sampling was performed in summer (May–July) and in fall/winter of 2015. In summer, only juveniles, having little prior chance to disperse beyond their natal habitat, were collected to establish fingerprints of source populations. Thus, their bone tissue will reflect the elemental fingerprint of the sites where they were collected. To make our trace element fingerprint library more representative, juveniles were collected from almost all the Okanagan-Similkameen cities/towns, and lands used for different agricultural activities, such as vineyards, orchards, dairy farms, feedlots, grasslands, and landfill sites.

In fall, problem birds (both adult and juveniles (<1 year)) were collected in vineyard/orchards and dairy farms/feedlots. The age of the birds was determined from their plumage (*Szép et al., 2003*; *Szép et al., 2009*). Additionally, aging by plumage was validated by the chronology of skull development (skull ossification–skulling) in juveniles (*McKinney, 2004*). Five randomly selected samples, which were considered as juvenile through plumage, were dissected to observe developmental stage by skulling (*Mueller & Weise, 1996*). These starlings are considered pests and hereafter referred to as "problem birds".

Sampling was performed in collaboration with the Starling Control Program (SCP), British Columbia Grapesgrowers' Association (BCGA). The work was conducted in accordance with the ethics training requirements of the Canadian Council on Animal Care (CCAC)/ National Institute Animal User Training (NIAUT) program certificate number 7252-1. A team of professional trappers contracted by the BCGA, has been trapping and euthanizing starlings following the guidelines of the Canadian Council on Animal Care every year. We obtained frozen sample from SCP trappers and each individual bird is considered as a single sample. Altogether, 105 juveniles were collected from ten different locations situated at distances from 12 to 190 km apart throughout the Okanagan-Similkameen region in 2015. The number of juvenile samples collected in summer varied depending on availability at each location, 20 from Kelowna, 10 from Hullcar, 10 from Salmon Arm, 9 from Armstrong, 9 from Mara, 18 from Vernon, 10 from Oliver, 8 from Osoyoos, 6 from Penticton and 5 from Keremeos. In fall, 118 problem birds (including young adults <1 year, $n = 20$; and adults: 1+ years, $n = 98$) were collected from vineyards/orchards and dairy farms and feedlots having major starling problem. Some of the problem bird sampling locations are the same where juveniles were collected like Vernon ($n = 4$), Pentiction ($n = 27$), Keremeos ($n = 5$), Kelowna ($n = 14$), and Oliver ($n = 15$); however, Okanagan falls ($n = 4$), Summerland ($n = 30$), Coldstream ($n = 10$), and Lumby ($n = 9$) were added in fall. These locations spread across Okanagan-Similkameen region from south to north (Fig. 1).

## Sample preparation

Samples were prepared by methods described by *Norris et al. (2007)* with some modification. Briefly, the bone was cleaned by removing outer skin and marrow. Bones were washed repeatedly with ultra-pure water (18 M $\Omega$ cm$^{-1}$), dried at room temperature (20 °C) (*Szép et al., 2009*) and weighed. The clean fragments of the bone sample, weighing between 30 mg to 80 mg each, were put separately into Teflon tubes. Trace element grade

concentrated nitric acid (2 ml) was added to each tube and placed in a heating block 75 °C (±5 °C) until the bone was completely dissolved. Samples were then cooled to room temperature, and then one milliliter of trace metal grade hydrogen peroxide was added to further digest the dissolved organic compounds (*Norris et al., 2007*). Samples were then evaporated on a hot plate at 85 °C (±5 °C) (*Szép et al., 2003*; *Neuhauser, 2013*). After cooling, 2 ml of 1% $HNO_3$ with 1 ppb indium internal standard was added to each vial to dissolve the residue over a period of 2 h at room temperature.

Samples were transferred quantitatively with three washes of the $HNO_3$ indium solution into acid-washed high-density polyethylene tubes to a final volume of 10 mL (*Norris et al., 2007*; *Neuhauser, 2013*). Sample blanks were prepared in the same way for every set of digestions to monitor contamination (*Donovan et al., 2006*; *Norris et al., 2007*; *Szép et al., 2009*). Sub-samples were further diluted 100-fold with 1 ppb indium-1% $HNO_3$ for ICP-MS analysis and 5 and 10 fold with 1ppm yttrium-1% $HNO_3$ for ICP-OES analysis. Samples were diluted to fall within the detection limits of the instruments.

## Trace element analysis

We measured 21 elements: aluminium (Al), silver (Ag), barium (Ba), calcium (Ca), cadmium (Cd), cobalt (Co), chromium (Cr), copper (Cu), molybdenum (Mo), manganese (Mn), lead (Pb), sulphur (S), scandium (Sc), selenium (Se), tin (Sn), strontium (Sr), vanadium (V), zinc (Zn), magnesium (Mg), sodium (Na), and potassium (K) in bone tissue. The elements were selected on the basis that they are present at levels high enough for reliable quantification, and could also vary in space.

Seventeen of the 21 elements were analysed by Thermo-Fisher Element XR sector field Inductively Coupled Plasma-Mass Spectrometer (ICP-MS) (*Poesel et al., 2008*; *Neuhauser, 2013*). A subset of four highly-abundant elements, Ca, Mn, Na, and K, were analysed via a Thermo-Electron Corporation, iCAP 6000 Series XR Inductively Coupled Plasma-Optical Emission Spectrometer (ICP-OES) (*Szép et al., 2003*; *Szép et al., 2009*; *Neuhauser, 2013*). For ICP-MS, elements were analysed in either low resolution or medium resolution mode to resolve polyatomic interferences. Both instruments were calibrated by external multi-element standards. Four different concentrations of multi-element standards were run to produce calibration curves (*Poesel et al., 2008*). Indium (*Norris et al., 2007*) and yttrium (*Neuhauser, 2013*) were used as an internal standard for ICP-MS and ICP-OES respectively to correct for instrument drift, sample density and to improve accuracy and repeatability. The blank and reference solutions were run at 20-sample intervals throughout the analysis. Three replicate analyses were done for each sample. Relative Standard Deviation (RSD) of the analysis was between 1–2%.

## Statistical analysis of data

The elemental concentration was normalized to the concentration of calcium rather than mass to minimize variability in the ratio of mineral:collagen content. The collagen structure of bone bonds relatively poorly with metals, whereas the mineral apatite substitutes cations and anions relatively easily into its structure. As birds develop, the relative content of apatite increases (*Rogers & Zioupos, 1999*; *Currey, 2004*; *Currey, Brear & Zioupos, 2004*;

*Pasteris, Wopenka & Valsami-Jones, 2008*). Normalizing with calcium, therefore, removes this variability due to collagen content. After calcium normalization, the data were then standardized via z-scoring to prevent individual elements from having a disproportionate influence on the groupings derived from multivariate analyses (*Fowler, Cohen & Jarvis, 1998*; *Norusis, 2016*).

A multivariate analysis of variance (MANOVA) was used to analyse trace element profiles of juvenile bone tissue collected in different locations (*Norusis, 2016*). A multiple discriminant analysis (*Hair et al., 2010*) of juvenile samples was performed to discard elements (variables), which were little related to group distinction, and also to develop the predictive model of group membership based on trace elements (chemical profile of bone). A discriminant function (equation) was developed based on the linear combination of the predictor variables (trace elements) that provides the best discrimination (correctly separating individual birds) among the predetermined groups (*Lachenbruch & Goldstein, 1979*; *Greenough, Longerich & Jackson, 1997*; *Szép et al., 2009*). The predetermined groups were the sites where the juvenile starlings were caught. The stepwise method and Mahalanobis distance (*Hair et al., 2010*) were used for the analysis because of the large number of variables (20 different element concentrations).

The stepwise method looks at each element (variable), one at a time, and determines which element is the best predictor of group membership, and ultimately generates the best set of variables to predict group membership. Cross-validation assessed the success of the proper bird groupings via the discriminant function and rules. In order to classify objectively the problem birds with respect to origin, the value obtained from each function for each problem bird was normalized (ratio of sample/ratio of centroid) by the centroid value of each site. Thus, if an individual problem bird were to lie at the centroid for a particular site, its value with respect to that site would be one. Each bird's probable origin was identified by how closely that bird's normalized value approximated the centroid values of each site. For some of the problem birds, the discriminant function value was too high to assign a specific juvenile source population. For such birds, we have categorized them as immigrant/unassigned birds.

Cluster analysis (*Hartigan, 1975*) was performed to evaluate the spatial separation of the elemental fingerprint of starlings and the identification of likely immigrants. In this study, the ward.D method (*Ward, 1963*; *Murtagh & Legendre, 2014*) with Euclidean distance measures was used for analysis, which provided better separation of individual birds with similar signatures and geographic location. Additionally, an average method with correlation distance was used to compute a test of significance of the clusters (by calculating the *p*-value of the cluster). The R package "*pvclust*" (*Suzuki & Shimodaira, 2014*) was used; it uses bootstrap resampling techniques to compute the *p*-value for each cluster. This method generates thousands of bootstrap replications by randomly sampling elements of the data. For each of the clusters, Approximately Unbiased (AU) and Bootstrap Probability (BP) values were calculated. The AU probability values (*p*-values) are computed by multiscale bootstrap resampling where AU $\geq$ 95% are considered to be strongly supported by the data, while BP corresponds to the frequency that the cluster is identified in bootstrap copies (*Suzuki & Shimodaira, 2014*).

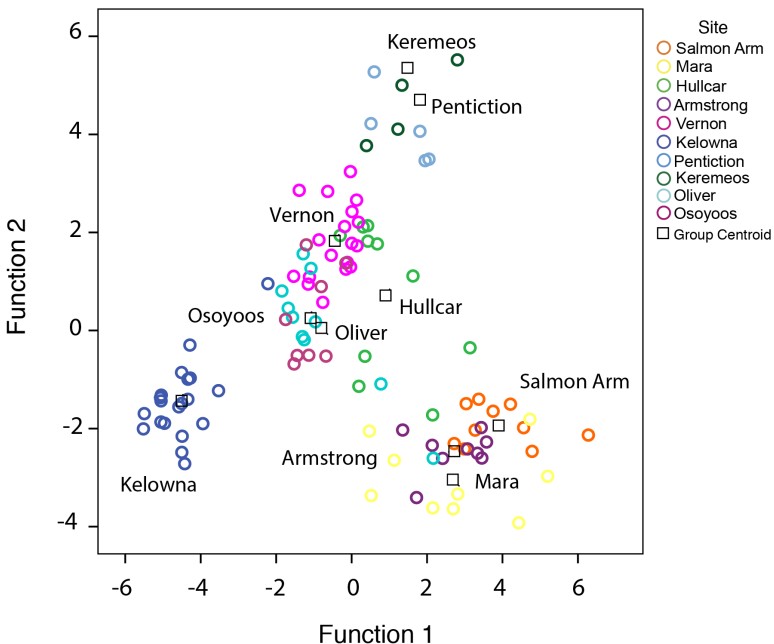

**Figure 2 Distribution of juvenile starlings caught in different locations in the Okanagan-Similkameen region, British Columbia.** Stepwise discriminant analysis of juvenile starlings ($n = 105$) from 10 different locations based on first two canonical discriminant functions ($rc = 0.07, P < 0.0001$) calculated from 20 trace elements. Different color represents different sampling area, and the black square represents the centroid of each sampling area.

Statistical analysis was conducted in SPSS ver. 24.0 and R statistical software version 3.3.1 (*R Development Core Team, 2014*) with the *cluster* (*Maechler et al., 2017*) *and pvclust* (*Suzuki & Shimodaira, 2014*) packages. Map projections of data were done in ArcGIS 10.4.

## RESULTS

### Spatial separation of source population of starlings in the Okanagan-Similkameen region

The source population trace element fingerprint library for the Okanagan-Similkameen region was developed from juvenile starlings collected from 10 different locations. Multivariate comparison of 20 elements in the juvenile ($n = 105$) population using MANOVA shows there is a significant difference in the trace elemental composition of juvenile bone among locations (Wilks' $\lambda$ <0.0001, $F = 6.655$, $df = 180, 645.230$, power = 1, $p < 0.0001$).

A stepwise discriminant analysis of 20 trace elements based on the juveniles ($n = 105$) collected at 10 different locations, provided four canonical discriminant functions with significantly high eigenvalues (>1), and canonical correlations ($rc > 0.70$), that separated juvenile samples from respective sites (Wilks' $\lambda = 0.001, 0.11, 0.73$, $\chi^2 = 633, 428, 247$, $p < 0.0001$) (Fig. 2). Ten elements (K, Mg, Na, Ag, Cd, Ba, Sc, Cr, Cu, and Se) out of 20, were used as the best predictor variables in discriminating group membership. The probabilities of correctly classifying juveniles to their respective locations using stepwise

Khatri-Chhetri et al. (2020), *PeerJ*, DOI 10.7717/peerj.8962

**Table 1 Classification results for juvenile starlings to its predicted locations in the Okanagan-Similkameen region, British Columbia.** The predicted group membership shows the percentage of correctly classified samples, based on the cross-validation function in the SPSS software package where 79.0% of original grouped cases correctly classified.

| Year | Site | Predicted Group Membership (%) | | | | | | | | | | |
|------|------|---------|---------|------------|-----------|------|--------|--------|---------|------------------------|----------|--------------|
| | | Kelowna | Hullcar | Salmon Arm | Armstrong | Mara | Vernon | Oliver | Osoyoos | Penticton/ Summerland | Keremeos | No of sample |
| 2015 | Kelowna | **95** | 0 | 0 | 0 | 0 | 5 | 0 | 0 | 0 | 0 | 20 |
| | Hullcar | 0 | **90** | 0 | 0 | 0 | 0 | 0 | 10 | 0 | 0 | 10 |
| | Salmon Arm | 0 | 0 | **70** | 20 | 10 | 0 | 0 | 0 | 0 | 0 | 10 |
| | Armstrong | 0 | 0 | 0 | **78** | 22 | 0 | 0 | 0 | 0 | 0 | 9 |
| | Mara | 0 | 0 | 0 | 33 | **67** | 0 | 0 | 0 | 0 | 0 | 9 |
| | Vernon | 0 | 0 | 0 | 0 | 0 | **100** | 0 | 0 | 0 | 0 | 18 |
| | Oliver | 0 | 0 | 0 | 0 | 10 | 0 | **60** | 30 | 0 | 0 | 10 |
| | Osoyoos | 0 | 0 | 0 | 0 | 0 | 37 | 37 | **25** | 0 | 0 | 8 |
| | Penticton | 0 | 0 | 0 | 0 | 0 | 0 | 0 | 0 | **67** | 33 | 6 |
| | Keremeos | 0 | 0 | 0 | 0 | 0 | 0 | 0 | 0 | 0 | **100** | 5 |
discriminant analysis are shown in Table 1. The table shows the predicted classification (group membership) using cross-validation from different locations compared to the apparent or actual observation. The overall probability of correctly classifying juveniles was 79% ($P < 0.0001$, Press's Q; Table 1). The juveniles were grouped 67%–100% correctly, with Osoyoos as an exception. Only 25% of the Osoyoos juveniles were correctly grouped by location. Mostly the misclassification pattern that occurred was among nearby locations. For example, a high rate of misclassification, 30% and 37% (Table 1), occurred between Oliver and Osoyoos (∼17 km apart), 33% between Penticton and Keremeos (∼45 km apart), and 22% and 33% between Mara and Armstrong (Table 1). Hence, almost all the misclassification occurs between sites that are very close to each other, with one exception, between Osoyoos and Vernon in 2015 (37%, ∼165 km apart, Table 1).

Cluster analysis of juveniles shows broadly similar results. The clusters (ward.D method) grouped juveniles from different sites into different clusters, even separating individuals derived from locations ≤12 km (Fig. 3) apart. Since there is no particular rule for choosing the height in the ward.D method dendrogram, the rule of thumb height of 10 was applied. Between 33–100% of juveniles from the same locations were grouped together at the height of 10 (Fig. 3). Approximately unbiased (AU) values obtained via the average method were used to test the significance of the clusters. Ten major significant clusters with red boxes (AU $p$-value > 0.95) separated juveniles from different geographic locations (Fig. S2). Thus, juvenile grouping with respect to geographic location in cluster analysis supports the predicted membership results derived from stepwise discriminant analysis. Thus, our results are robust to different multivariate statistical methods.

## Identification of the source population of problem birds in the Okanagan-Similkameen region

The source population of problem birds was identified by comparing their trace element fingerprint with the juvenile fingerprint library. Cluster analysis of all birds including both juvenile ($n = 105$), and problem birds (<1 year adults ($n = 20$), and 1 + year adults ($n = 98$)) depicts two distinct clusters, separating most problem birds into one cluster and another cluster comprising a mix of the remaining problem birds and juvenile birds (Fig. 4). Around half of the problem birds ($n = 52$, 44%) clustered separately, indicating chemical composition distinct from the juvenile fingerprint library (Fig. 4). Problem birds having trace element fingerprints distinct from the Okanagan-Similkameen populations (doesn't match our library) were considered as immigrant/unassigned birds, likely originating from unsampled sites within the Valley or from outside of the Valley.

To identify the origin of the remaining ($n = 66$, 56%) local problem birds in the Okanagan-Similkameen region, the same four canonical discriminant functions were used. The first function explained 44.2% of the total variance among locations. The second function explained 33.0%, and the third and fourth explained 11.7% and 6.1%, respectively of the total variance among locations. The problem birds collected in fall were divided into two groups. First, problem birds collected in vineyards and orchards and second, those collected at dairy farms and feed lots. The majority (55%) of problem birds caught in vineyards and orchards were consistent with trace element signatures of juvenile birds from

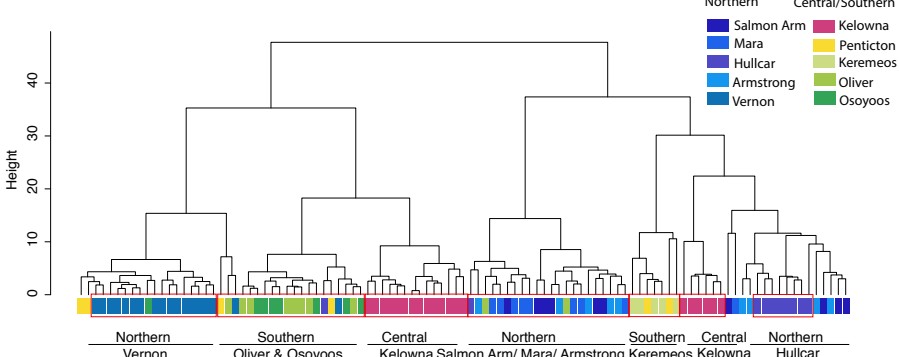

**Figure 3  Cluster analysis of juvenile starlings.** Cluster analysis of juvenile starlings ($n = 105$) sampled in the Okanagan-Similkameen region based on ward.D method derived from the Euclidean distance using 20 elements. The distance provides a relative measure of how different the clusters are in relation to each other. Different colour represents different locations in the Okanagan-Similkameen region, British Columbia.

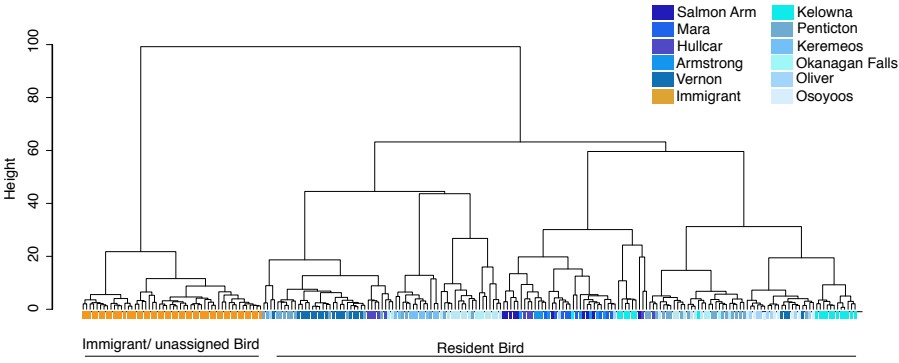

**Figure 4  Cluster analysis of juvenile and adult starlings.** Cluster analysis of juvenile and adult ($n = 223$) birds collected in different locations in the Okanagan-Similkameen region, British Columbia based on ward.D method derived from the Euclidean distance using 20 elements. Orange color represents immigrant adult adults, dark blue colours represent samples from the northernmost area, and the colour fades as the location moves towards the south.

the North Okanagan. Specifically, the highest contributing site was Vernon (41%) (Fig. 5). The remaining 45% of problem birds were a mixture of local and immigrant/unassigned birds (29% southern local, 2% central local), 14% of problem birds in vineyards and orchards were immigrants/unassigned (Fig. 5). In contrast to vineyards and orchards, the problem birds caught on dairy farms and feedlots were largely immigrants/unassigned (89%) and only 11% were local from northern site (Vernon) (Fig. 5).

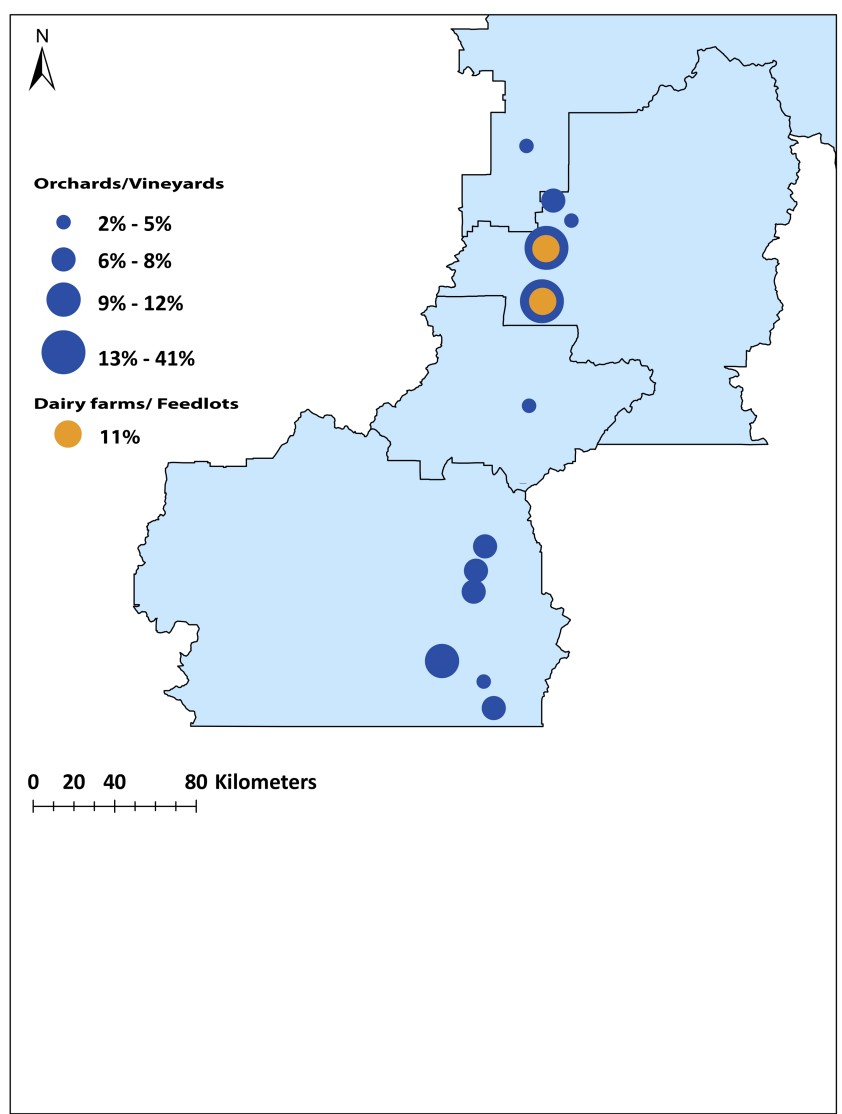

**Figure 5** **Map of the origin of the source population of problem birds caught in vineyards /orchards and dairy farms/feedlots in the Okanagan-Similkameen region, British Columbia.** The radius of the circle depends on the percentage contributed from particular sites.

## DISCUSSION

### Spatial separation of starlings

Resolving populations in space revealed distinct trace elemental signatures within the 200 km length of the Okanagan-Similkameen region. The spatial separation of starling in this study using bone tissue trace elemental fingerprinting is comparable to spatial separation of other birds at scales ranging between kilometers to continental distances using feather tissue trace elemental fingerprinting (*Poesel et al., 2008*; *Szép et al., 2009*). For example, white-crowned sparrow (*Zonotrichia leucophrys* F.) samples collected from four dialect populations were distinguished within 400 km (*Poesel et al., 2008*). Similarly,

sand martin (*Riparia riparia* L.) and barn swallow (*Hirundo rustica* L.) populations were separated within and between continents (i.e., across Europe and between European and African moulting sites) (*Szép et al., 2003*; *Szép et al., 2009*). In our study, the shortest distance separating sites with distinct chemical signatures was 12 km (between Vernon and Armstrong).

Site-specificity of the elemental compositions of bones from juvenile starling is similar to analyses of natal origin from nestlings and juvenile sand martins (*Riparia riparia* L.) (*Szép et al., 2003*) and for western sandpiper using feather tissue (*Calidris mauri* Cabanis; *Norris et al., 2007*).

Moreover, a high degree of similarity in feeding and foraging habitat of juveniles could be factors maintaining the similarity of juvenile bone trace element composition in local populations. Since most juveniles collected within sites were chemically similar, it appears that these juveniles had not dispersed. Thus, the juvenile fingerprint library adequately represents the fingerprints of source populations at particular sites in the valley.

The chemical signature of a few juvenile birds caught within a site matched with fingerprints from other sites according to the predicted group membership derived from discriminant analysis. Such misclassification of juveniles among nearby sites such as Oliver and Osoyoos may have arisen due to overlapping foraging ranges of the juveniles or adults feeding chicks, geological similarity between nearby sites, similar agricultural practices and food choice. These mechanisms are not mutually exclusive and all may have been operating. Finally, misclassification noted in a few birds between the sites that are 100s of kilometers apart geographically might be due to differences in food choice and early short distance dispersal or geochemical similarity between remote sites.

The overall degree of classification of juvenile fingerprints was 79% among sites. This is similar to the degree of differentiation reported for shearwaters (*Calonectris* Cory & Oustalet) 75-89.9% (*Gómez-Díaz & González-Solís, 2007*) and lower than that in white-crowned sparrows 100% across 400 km (*Poesel et al., 2008*). The dissimilarity of chemical composition among sites within the Okanagan-Similkameen region may be due to geologic factors (*Ethier et al., 2013*), and/or hydrologic factors including irrigation (*Poesel et al., 2008*). The underlying geology of the Okanagan-Similkameen watershed is very diverse within small spatial scales (*Okulitch, 2013*).

Similarly, soil chemistry is non-uniform over the landscape (*Goitom Asfaha et al., 2011*). Water, mineralogy and soil microbiology affect soil chemistry. Solutes present in soil water can be derived from weathering of rocks and minerals locally or could have come from water (precipitation and irrigation). The trace elements move from minerals to soil water (*Galgano et al., 2008*) where they are taken up by plants. In turn, plants form the base of the food web, eventually accumulating in animal tissues, forming a unique marker. The trace element availability in soil can be further modified by factors like soil pH, water content and porosity (*Kelly, Heaton & Hoogewerff, 2005*; *Kim & Thornton, 1993*).

Trace element composition in bone tissues can yield a very specific biogeochemical marker. Out of 21 elements analysed, ten elements (K, Mg, Na, Ag, Cd, Ba, Sc, Cr, Cu, and Se) were used in a discriminant analysis that classified juveniles source populations. All are cations (Se is also an oxyanion) and could potentially substitute into apatite. For

example, Pb substitutes for Ca in hydroxyapatite in bone with a turnover rate ranging from 1–8% per year depending on the type of bone tissue (*Martin et al., 2007*). The shortlisted elements from the discriminant analysis are therefore a mixture of those elements, which could easily be present in bone, and could also vary in space.

## Assigning problem starlings to natal populations

Problem birds were assigned to potential natal populations through discriminant analyses. However, some of the problem birds could not be assigned to its source population and were designated as immigrant/unassigned problem birds. Failure to assign origins for a portion of the problem birds could be caused by continued addition of elements to bone over large unknown space over time. Second, adult starlings >1 year old, natal fingerprints may differ inter annually, especially if foraging by such a generalist might change for different climatic years. Third, problem birds could come from outside the region.

The problem bird contribution to vineyards and orchards in the southern part of the Okanagan-Similkameen region was mostly from the northern end of the valley. It is likely that the birds reared in the northern part of the valley move south as fruit ripens in the fall. The remaining problem birds in vineyards and orchards are a mixture of local (southern) and immigrant/unassigned birds. This is consistent with documented migration patterns in starling. Generally, fall migration occurs from September to December (*Kessel, 1953*; *Dolbeer, 1982*).

The problem birds on dairy farms and feedlots were a mixture of immigrants/unassigned and local birds. The high immigrant/unassigned population in the north Okanagan is likely because these sites are close to unsampled source populations further to the north. During winter, dairy farms and feedlots provide, adequate shelter and warmth and a diverse supply of food and water (*Palmer, 1976*). Several past studies have noted that the flocks of starlings increase in number and concentrate on dairy farms and feedlots in winter (*Palmer, 1976*; *Glahn, 1981*). Moreover, one of the challenges to this type of studies is that the trace element profiling is more difficult for wide-ranging, continuously distributed species as the number of potential source areas is vast. If the organism migrates long distances, then trace elements alone might not be sufficient to track their origin. Also trace elements alone would not be sufficient for tracking the origin of migratory organisms that undertake large-scale movements and forage in diverse geochemical environments. However, the combined use of trace elements and isotopes could help track continental-scale movement patterns. Besides identifying bird origins, the trace elemental signature could be used in several other ecological studies of birds, including matching breeding populations with overwintering populations, tracking fine-scale dispersal (*Poesel et al., 2008*), identifying migratory connectivity (*Szép et al., 2009*) and identifying moulting areas (*Szép et al., 2003*).

## CONCLUSION

The results provide evidence that bone tissue elemental fingerprints of migratory pest starlings can be used to identify the natal origin of starlings with a high spatial resolution (10s of km). The technique can be very effective in an area like the Okanagan-Similkameen region with highly diverse geology. Most of the problem birds in vineyards and orchards of

the south were identified as local birds; very few were immigrants/unassigned. In contrast, in dairy farms and feedlots in the north, most of the problem birds were identified as immigrants/unassigned to the valley, probably because migration in fall/winter is generally southerly and the populations to the north were not sampled. This study reveals that trace element fingerprints, a natural environmental tracer, can track the origin of mobile organisms like starlings. Knowing the origins of problem birds enables managers to make more informed decisions about how and where to concentrate control efforts.

## ACKNOWLEDGEMENTS

We would like to thank Mr. David Arkinstall for his support in the trace analysis lab. Thanks to Ms. Ashleigh Duffy, Mr. Kevin Kuemper and Dinesh Adhikary for their support during field trips.

### Funding

This work was funded by British Columbia Grapegrowers' Association and Mitacs. Additionally, this project was funded through the Agriculture Environment Initiative with funds sourced from Agriculture and Agri-Food Canada and the British Columbia Ministry of Agriculture, and administered by the Investment Agriculture Foundation of British Columbia. There was no additional external funding received for this study. The funders had no role in study design, data collection and analysis, decision to publish, or preparation of the manuscript.

### Grant Disclosures

The following grant information was disclosed by the authors:
British Columbia Grapegrowers' Association and Mitacs.
Agriculture and Agri-Food Canada.
British Columbia Ministry of Agriculture.
Investment Agriculture Foundation of British Columbia.

### Competing Interests

The authors declare there are no competing interests.

### Author Contributions

- Upama Khatri-Chhetri performed the experiments, analyzed the data, prepared figures and/or tables, authored or reviewed drafts of the paper, and approved the final draft.
- John G. Woods and Ian R. Walker analyzed the data, authored or reviewed drafts of the paper, and approved the final draft.
- P. Jeff Curtis conceived and designed the experiments, authored or reviewed drafts of the paper, and approved the final draft.

### Data Availability

The raw data is available in the Supplementary Files.

## Supplemental Information

Supplemental information for this article can be found online at http://dx.doi.org/10.7717/peerj.8962#supplemental-information.

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
