# Peer review of "Origin identification of migratory pests (European Starling) using geochemical fingerprinting"

_PeerJ, doi:10.7717/peerj.8962_

## Round 0.1 · original submission · Major Revisions

Thank you for submission and effort. It was well reviewed and overall the comments are constructive and promising to revise and resubmit. I am happy to see a revision and, if comments are addressed, accept this manuscript for publication.

One note, please try and expand on any data that might characterize the underlying geology of sites. This is a limitation you address in the paper, but you still broadly make statements that assume you have better site characterization of underlying geology. Please try to address this, while addressing the comments from reviewers. I think the reviewers have provided an excellent road map to revise your work and the detail of the reviews confirms it is an interesting paper. Please make a full faith effort to revise and resubmit.

Nice work.

·

Basic reporting

Generally clearly written, though some minor issues with sentence structure and grammar exist.
There are some places in the intro that could use a bit more explanation (see general comments section). For example, more could be presented about the known migration patterns and distances for starlings relative to the study region and site delineations. But overall the context is clear and the intro has a logical flow.
There are numerous examples where statements are made without any citation (see general comments section).
The figures are generally relevant and of high quality. However, I have a couple of suggestions for improvements (see general comments section for details). For example, Figure 1 would be a lot more informative with more information about underlying geology dictating the regional trace element landscape. And all three figures could use more clear indications of site locations to help orient us in spatial distribution.
The Zscored trace element data are present in the raw data file but I don’t see the original ICP-MS and ICP-OES data.

Experimental design

This manuscript certainly reflects original primary research within the scope of PeerJ. I think the research question is clear, well defined, and interesting/valuable. The rationale for need is generally pretty clear, though it seems as though the authors were simultaneously arguing that little is done with trace element fingerprinting in bones so we don’t know if it works (not really true as trace elements in bones have been use for some time now to look at geographic origin in a wide range of species) and also applying it with the assumption that it does work.
The methods could be a bit clearer. In particular, there is very little information about the regional geology that establishes the geospatial patterns in trace element profiles. It would be useful to see that on the figure and discussed in the “Study Area” section. It could even be used to make hypotheses about site separation that could be tested with the data presented.
Little information was presented about the birds sampled (size, sex, age etc.) that could be used to help interpret the patterns in the context of movement ecology.

Validity of the findings

I think the authors did a nice job interpreting their data, but there are several important assumptions made that I think need to be clearly identified and discussed (see general comments for details). My biggest issue is that it is really challenging to be certain of site identifications if you do not have all potential sources clearly characterized. The classification model is only as good as the data you put in. So if you only have a subset of the sites characterized, the model will do look to assign groups to those sites. You clearly showed that there can be significant variability on short spatial scales within your study region. So why couldn’t that same variability exist outside the study area such that there are other regions with similar geology to your study sites? Similarly, in L366-371 you argue that some birds could not be assigned to origin sites because they are from areas between the sampling locations and trace element signatures varied over small spatial scales. I feel like this line of reasoning is exactly why it is challenging to say whether the birds that had values distinct from your juvenile source locations were immigrants or not. They could just reflect local sites not sampled.
I’m also interested to know how stable are the geospatial trace element signals. The juveniles representing source locations were collected in one season and year, yet the adults were collected over more than a decade. How variable are the source signals seasonally/interannually relative to spatial across the different potential origin habitats?
The authors often discuss the results providing “high spatial resolution” over “small spatial scales” but these are very qualitative metrics without reference to the movement scales and habitat scales of the study system. Small scale is very different for a starling vs a wandering albatross. So unqualified statements like this are a bit empty. Similarly, statements like “The selected trace elements may provide the best marker for tracing the origins of starlings and similar birds.” (L375-376) are not really accurate. The elements selected are specific to your study region and underlying geology. They are not universally the best. A region with different underlying geology will likely require a different suite of elements to properly characterize those sites.

Additional comments

L2: No need to include an abbreviation for European staling if you don’t use it again in the abstract

L6 etc.: Some issues with comma usage need to be addressed. E.g., No comma after contaminating food (L6). Include a comma after (Cabe 1993) (L28). Include a comma after parasites (L33). Comma after In 2003 (L52), and many more

L13-14: I’d suggest rephrasing to “…classified juveniles to their natal origin (minimum discrimination distance of 12 km) with 79% accuracy.”

L14-17: I’m a little confused here. Is this saying that 11% were local and 89% were not local, and of the 89%, 55% were from the northern portion of the valley or 55% of all birds were from the northern portion of the valley?

L18-20: How did you determine the spatial accuracy? You would need independent validation to determine that the birds actually came from those locations (e.g., tagging work coupled with trace element work). Also, how do you quantify “high degree” and “small geographic scales”? these are qualitative metrics relative to the organism and question at hand. Small geographic scales for a non-migrant mean very different things than small scales of a global migrant. This could use a bit more explanation, e.g., high degree of accuracy relative to typical migration distances etc.

L22: When were European starlings introduced? Is this a new threat or something that has been here for decades or centuries?

L24: Is the IUCN status of least concern the metric of consideration for it being one of the most successful non-native species?

L26-36: These five reasons seem to lack some key citation. This first example seems a bit vague, citing a government report on starlings. What do you mean by adaptive in nature? Is this all linked to being a generalist forager? Where is the evidence of them being resilient and adaptive to extreme and diverse habitat conditions (no citations here)? Some added citations clearly assessing their resilience and adaptation would be useful here. Similarly, there’s no citation at all for the broad claim that predators, parasites, and diseases are not sufficient to inhibit populations growth. There’s also no citation for the North American population size.

L37: You introduced an acronym EUST in the first sentence but then don’t appear to use it. Please be consistent here. I personally prefer spelling the name out. EUST does not seem to be standard and the abbreviation really doesn't save much space.

37-47: There is a lot of information here that seems like it should have citations. Where is the quantitative evidence that starlings cause significant financial loss to farmers in the region? Similarly, there is no citation for “large flocks destroy fruits…” Since this paper rests on the notion that these starlings are significant pests, the sources of this information should be documented.

L47-49: Are these figures for damage caused by European starlings? If so, state that. If not, then this sentence is a bit misleading and should be modified to reflect that.

L50: Propane cannons!?! I’ve never heard of those but it sounds crazy.

L54: Citation for this statement about industries supporting the program.

L57-59: Again, to make a claim that numbers of starlings are increasing, you need to have a citation that shows quantitative evidence.

L67-70: These techniques are definitively potential techniques, but they aren’t the only techniques. People use population genetics, tagging/tracking techniques etc. I suggest removing “the” from in front of potential methods to make it clear that these methods are the only options used. Similarly, the markers can be indicators of origin, but they aren’t always. Plenty of isotopes record information about things other than origin (diet, trophic dynamics, metabolism etc.) and there are plenty of times where isotopes and trace elements don’t clearly distinguish origins because the sites do not have different signatures. This is all just a case of phrasing (easy fix), but makes a big difference in the intent and interpretation of the statements.

L71: There is very little discussion about the known movement patterns of starlings. How far do they travel and over what time scales? Do they undergo daily migrations, seasonal migrations, ontogenetic migrations? Do they return to their natal origin repeatedly or do they leave and not return?

L74-75: Why not use both? Isotopes and trace elements record different mechanisms of variation, which can complement each other and improve accuracy.

L84: Conversely, the bone also integrates over the lifetime of the bird. So if the bird is several years old, the vast majority of the bone signal will reflect life outside the natal origin. How do you control for the relative input of signal in bones? The older the bird gets, the more diluted the natal signal.

L91-93: What was the metric for determining “the best”? Accuracy, precision, cost, time, relative ease?

L93-94: I think this section needs a lot more attention. The entire study hinges on solid geographic separation of source regions based on trace element profiles. So there should be a clear discussion of the patterns and drivers of distribution, the stability of the signals, the spatial scales of separation relative to starling migration patterns etc.

L104-115: I’m struggling a bit to figure out why this information is really critical for the study area description? This seems like an expansion of background info already presented in the intro, but not directly relevant to the methods of the study. More useful information here would be the shape of the valley and the distribution of habitats in relation to bird migration, discussion of the bedrock distribution and resulting geospatial trace element signals etc. These info are critical to evaluating the use of this fingerprinting technique for tracking movement patterns.

L120-121: This sentence is a bit awkward. Consider rephrasing to “We obtained fresh and frozen samples, with each bird representing a single sample.”

L124-125: This technique needs a citation.

L127: What does “shortly after”

L117-131: I’m a bit confused by this section. The topics jump back and forth between collecting adults and juveniles. I would recommend reorganizing this a bit. First the sentences about juveniles representing local source signals needs to be moved to the beginning of the paragraph before talking about how many were collected and where. I would then talk about collecting since they are being compared to the juvenile source signal. I’m a little confused about the difference between the adults collected by SCP since 2003 and the “problem birds” collected in the Valley in 2015. Where were the rest of the adults collected? What makes a bird a “problem bird”? What are the sample sizes from the SCP and what are the total sample sizes for juveniles and adults. How many regions did you sample and how did you define those regions (by habitat, by cardinal location in the valley, by distinct trace element signals)? I’d like this section to be a bit clearer and straight forward as to how many birds were collected, what life stages, what regions, and what years.

L121-123: How stable are the geospatial trace element signals. The juveniles representing source locations were collected in one season and year, yet the adults were collected over more than a decade. How confident are you in the accuracy of your source location signals? In aquatic systems, terrestrial bedrock provides the trace element signal in rivers, which get recorded in anadromous fishes. To look at natal origins, scientists have to ground truth the source signal to the season and year in which the fish occupied that habitat (e.g., characterize the source signal in 2015 and look at one year old fish in 2016 or two year old fish in 2017 etc.). The source signal will change not only as a function of bedrock type (very stable), but also temperature, precipitation, growth conditions of plants taking up the mineral signal etc. (much less stable). How variable are the source signals seasonally/interannually relative to spatial across the different potential origin habitats?

L134-137: I know I’ve been harping on adding citations, but I don’t think you need multiple citations for each step for standard protocols like drying to room temp or weighing or putting a sample in a Teflon tube. At best include the citations at the end of the sentence, but honestly I would just cite steps that are unique to this protocol since you already cited Norris et al. 2007.

L157-160: What about selection based on what elements help discriminate geographic regions? The Szep et al. 2009 paper cited as evidence of using the elements to look at migration was working Europe and western Africa. There is no reason to believe that those same elements should be the best choice for western Canada. You should be able to determine the predicted elemental composition based on bedrock maps of the region. This would be a much better justification for use.

L184: Trace Element Multivariate Analysis is really a component of the Statistical Analysis of Data and should be included in that subsection.

L193: How did you determine “site”? There was never really and discussion of what qualifies as a site beyond the southern fruit and northern dairy portions of the valley.

L223: Here and throughout, it would be useful to more clearly differentiate headings and subheadings. E.g.,
Results
Spatial separation of source population of starlings in the Okanagan Valley

L226: I don’t think the Results section should be the first time we are told that there are 10 potential juvenile source locations. There should be a much clearer discussion earlier in the methods about what those sites are and how they were determined.

L230-232: I don’t think a single sentence should be a stand-alone paragraph.

L258-259: This is nice.

L261: How do you define “immigrant”? Is it a distance away or simply that they don’t match the juvenile source signals? Could there be regions within the valley that were not characterized?

L278-280: This is not a proper sentence “Out of which…”

L286-287: First off, there’s no citation for this statement that results from your study were comparable to feather-based studies. Second, I’m not really sure what this means. The statement is a bit vague. In what ways were they comparable? The absolute distances, the distances relative to bird movement scales, the distances relative to geographic bedrock variation?

L287-293: Local/regional scales is very subjective and will depend on the species. That said, between continents seems larger than local to regional scales.

L295-297: Again, I’m a bit confused by this comparison. The exact distances are case specific. The spatial scale will depend on the geographic composition of bedrock, which is site specific, and whether or not that separation scale is useful will depend on the species. For example, one site might change bedrock type on the scale of kms and the other 100s of km. Similarly, for a bird that migrates 100s of km, separation at the scale of 10s of km might be great, but if the bird travels kms, then separation at the scale of 10s of km isn’t very helpful.

L301-304: I feel like there is plenty of evidence for this. These phrases make it sound like this was in question.

L305-307: I’m quite confused by this sentence. First, I’m trying to figure out what it adds to this discussion and second, I’m not sure what you mean by “phenotypic quality”.

L315-317: How old were the “juvenile” birds? The elemental signal in natal origin birds would reflect the diet of the adults feeding the them until they can feed on their own. When they are fed by the adults, their signal will reflect the spatial integration of the adult foraging range. How far to the adults travel to feed? In other words, how do the spatial scales of adult foraging compare to the spatial dimensions of the a priori sites selected? If adults forage in a radius beyond the local site level, then the “local signal” could actually reflect a much wider spatial scale that presented. Similarly, how long have the juveniles been feeding on their own, which would presumably reflect the more local signal without significant dispersal?

L325-328: 79% is not really an assessment of the degree of spatial separation or degree of differentiation, but rather an indication of reclassification accuracy.

L330-334: This information should be expanded upon. You could even turn this into hypotheses of site separation that you could test with your bird trace element profiles.

L336-342: This paragraph starts out saying the results can be compared to other studies, but then it doesn’t actually do that. I would suggest starting with an assessment of your results first, then talking about how your results compare to other studies, with specifics.

L346-348: The line “This implies that…” is repetitive of the sentences before.

L348-349: It would be helpful to have a clearer description of the known migration patterns in the intro to set up the context for interpreting the movement patterns determined from this study.

L353: How do you know that they were from Northern BC? I thought the evidence you had was just that the profiles didn’t match the juveniles you sampled.

L355-357: Why would a higher number of dairy farms lead to higher immigration? Are dairy farms more desirable than fruit farms?

L360: You've already spent much of the discussion talking about immigrants, so this definition here seems out of place.

L363-365: I don’t quite follow this line of reasoning. You clearly showed that there can be significant variability on short spatial scales within your study region. So why couldn’t that same variability exist outside the study area such that there are other regions with similar geology? There have to be maps of bedrock geology for the region which could be cross referenced with the signals from this study to make predictions about regions of similar trace element signal in areas within the migration range but outside the study area.

L366-371: I feel like this line of reasoning is exactly why it is challenging to say whether the birds that had values distinct from your juvenile source locations were immigrants or not. They could just reflect local sites not sampled.

L375-376: This is an odd statement. The elements selected are specific to your study region and underlying geology. They are not universally the best. A region with different underlying geology will likely require a different suite of elements to properly characterize those sites.

L377: Again, these are very qualitative metrics “small scale” and “high precision” without reference to the movement scales and habitat scales of the study system. Small scale is very different for a starling vs a wandering albatross. So unqualified statements like this are a bit empty.

Figure 1. Can you put the site names on the map to help orient us in the other Figures? The map would also be a lot more informative with more information about underlying geology dictating the regional trace element landscape. Outlining the spatial boundaries of the 10 collection sites and two major farming regions would also be helpful for context.

Figure 2. Can you order the names from North to South like in Fig 3 to again help orient to spatial distribution?

Figure 3: Can you adjust the color bar to provide a bit clearer distinction among the 10 sites. The all blue spectrum makes it really challenging to differentiate.

Reviewer 2 ·

Basic reporting

This paper presents geochemical fingerprinting as a novel approach to assessing the source of problem birds for damage management. The manuscript is well written. I have some minor grammatical comments, as well as some comments on the background and literature cited. One of the reasons the authors list as making starlings a highly successful pest is intraspecific brood parasitism. The paper cited (Kennedy and Harry 1999), has been incorrectly cited and should be Kennedy and Power 1990. In reading Kennedy and Power I do not see the support for the statement that the brood parasitism increases success rate and the number of chicks that fledge in starlings. I would suggest the authors remove this from the manuscript. The other four reasons for starlings being a “highly successful pest” are more than adequate. I find the authors reasoning for determining the natal origins of the pest birds to be flawed. The authors describe deterrent techniques “falconry, noise deterrents, electronic distress calls, bird netting, visual repellents) as eradication techniques. These techniques should be deployed at the orchards, dairies, and feedlots to prevent damage, not at breeding or origin sites. Some additional information on the lethal removal and where it is currently occurring that is not helping the damage situation would help strengthen this argument. A discussion of how deterrent techniques can be ineffective (and many of the techniques listed by the authors are ineffective except netting, which is effective, but can be prohibitively expensive) and that lethal removal, especially of a non-native invasive species can be beneficial would also strengthen the introduction.
Damage at orchards, vineyards, and berry production facilities by starlings has been well documented, and it is generally understood that there is an age difference in starlings that damage these locations. (Virgo 1971 “Bird damage to sweet cherries in the Niagara Peninsula, Ontario”; Stevens 1985 “Foraging success of adult and juvenile starlings Sturnus vulgaris: a tentative explanation for the preference of juveniles for cherries”; Conover et al. 2007 “Use of decoy traps to protect blueberries from juvenile European starlings”, etc…) This should maybe be pointed out in the introduction or discussion.
The structure conforms to the PeerJ standards, and the raw data has been supplied. The description of Figure 3 should include more specifics such as what distance are the authors referring to? The distance from one end of the cluster to the other, the distance between sites? Figure 4 I think would be more useful if it only included the adult birds. With the juveniles included as well the figure is small and hard to decipher. I find Figures 5 and 6 to be confusing and do not add much to the manuscript.

Experimental design

The research described in this manuscript is original and within the scope of the journal. The research question has been well defined, with the exceptions noted above. I think there could be some additional detail described in the methods. How were the sites for the adult birds chosen, how were the birds confirmed as being adult, especially with the timing of adult problem bird collections (Aug-Dec) and the timing of molt in starlings (Aug-Oct) (Birds of North America Online). There is literature suggesting that starlings disperse widely from their natal sites (Cabe 1999). This suggests that the authors methods will only be able to determine the location of juveniles born in the year the geochemical fingerprinting was conducted. Thus it is not surprising that 58% of the problem birds were immigrants or could not be assigned to a location.

Validity of the findings

No comment

Additional comments

L5: Replace “create” with “cause”
L7: Replace “deterrents” with “devices” or remove “scare”
L7: Remove “only”
L7-9: Do the authors have any literature to back this statement up? It typically requires more effort to trap birds before they begin to group into large flocks.
L37: “Starling(s) cause significant…”
L38: Although the location of the Okanagan Valley is defined in the abstract, it should also be defined here at first mention in text.
L41: BC is used with no description of what the abbreviation stands for.
L47: Is the damage described in US or Canadidan dollars?
L57: To put the bird removals in perspective it would help to have an estimate of the breeding population of starlings in the Okanagan Valley.
L59: Are the increased starling numbers specifically in the fall? Is it known how many EUST migrate into the valley in the fall?
L59-60: This logic does not make sense to me. The techniques described (falconry, noise deterrents, electronic distress calls, bird netting, visual repellents) cannot be applied at breeding locations and be expected to have an effect on damage at orchards and dairies. If the techniques implied here are lethal this should be specified.
L76: BC should be defined in L41.
L113-115: This statement was already included in the introduction, it should only be in one section and does not need to be repeated.
L266-267: Is it possible that these 44% are not immigrants? Starlings are known to disperse widely from their natal origins (see Cabe 1999 “Dispersal and population structure in the European starling”), thus these birds could be breeding in the Okanagan valley, but were born outside of the Okanagan valley. The method described in this paper would not be able to distinguish these birds from fall migrants.
L130-131: If problem birds were collected between August-December, how were they verified as adult birds? Juveniles molt in August-October and afterward look very similar to adult birds. Also, from how many sites and how were these spread around the valley. What proportion of the number of problem birds in the valley is the 118 bird collection and how representative of the problem bird population are these 118?
L441: Kennedy, E. Dale and Harry W. Power (1990).

·

Basic reporting

• Clear writing, logical flow, easy to read (thank you!)
• Introduction missing some references and details. See detailed comments and pdf markup.
• Figures 5 and 6 could be collapsed into one figure with different colour schemes
• Uncertain if the raw data were supplied

Experimental design

• Question well defined and fills a relevant knowledge gap
• Technical approach appropriate. I like how young birds were used to define the elemental origin of older birds!
• Ethical certificates identified
• Methods are detailed and reproducible

Validity of the findings

• Statistics are explained well and are appropriate
• Findings are discussed in the context of existing research
• Implications of finding outlined and linked to other research

Additional comments

Thank you for the opportunity to review this manuscript. It was a pleasure to read. My overall comments are minor (see pdf markup and comments below). I trust you can quickly address these to move this manuscript forward in the publication process.

Introduction: Overall very clear. The framing of the problem was well done. One thing that I would like to see added is a short description of the life cycle of Starlings in the area. For example, are the ‘problem’ birds migratory, stop-over, overwintering, ect? You talk about natal origin, which implies these birds are outside the breeding grounds? Or are the birds non-migratory or short-distant migrants? I know the answers to these questions, but a non-avian expert may not. Making this clear at the outset will better frame the problem for the non-expert. It would also be good to know if the birds show site fidelity to the breeding or wintering areas, or both? From a trace element perspective, I am curious how long do Starlings live? When are the trace element signatures suspected to have been incorporated into the bone tissue of adult birds? Is this a continuous process?

Conclusion: Be clear what the conservation implications are of these finding. For example, how will discerning natal origin help resource managers control problem Starlings? This is stated in the introduction but should be elaborated on here.

---

## Round 0.2 · Minor Revisions

Congratulations on a job well done in revising the manuscript in a way that appeases all reviewers. This is a nice revision and I fully expect to accept it once the minor comments of the three reviewers are addressed. Please make a quick round of edits addressing this minor comments. Thank you!

·

Basic reporting

Revisions make this component much improved. However, the paper needs a thorough read through for numerous grammatical issues.

Experimental design

no comment

Validity of the findings

I wish the paper could have better addressed the issues of uncharacterized sites, but I think they did a reasonable job acknowledging the uncertainty in their interpretations as a result.

Additional comments

I think the authors did a nice job addressing my earlier comments and suggestions. The addition of the table with geological information is quite helpful (bummer that there’s not a nice map of it). Thanks for including the original data in addition to the z-scored data. I appreciate the improved citation effort as well. It helps provide context for the material presented here, and maintains a paper-trail for the ideas presented by others. Overall, I think the manuscript clarity if much improved. However, the newly added text needs to be carefully reviewed for numerous writing clarity and grammatical issues. Nice job.


Specific Comments:
L86: Clarify large flocks of starlings

L148-9: I’m not a fan of the inclusion of a strawman hypothesis that we know will be rejected, but maybe that’s personal preference. I would prefer to see a more impactful “alternative” hypothesis, i.e., a statement of what you actually think you will see and why.

L474: Again, clarify starling populations

L556-563: The clarification of likely challenges/alternative explanations here is useful.

L557-558: I found this sentence confusing

Reviewer 2 ·

Basic reporting

This paper presents geochemical fingerprinting as a novel approach to assessing the source of problem birds for damage management. The manuscript is well written with the exception of some minor grammatical considerations (see below). The figures are relevant and add useful depictions of the results and locations.

Experimental design

The research described in this manuscript is original and within the scope of the journal. The research question has been well defined. The methods have been well described and with sufficient detail and information to replicate.

Validity of the findings

The authors conclusions are well stated and are reflected in the data presented.

Additional comments

L7: Replace “divert” with “disperse”; “acclimated” with “habituated”
L31: “…status is considered to be of least concerned…” Delete “ed” on concerned. The species classification is “least concern”
L39-40: “Starlings can migrate long distances in a day and can cover a total of up to 1,000-1,500 km over migration periods.” As written this sentence is misleading. Starlings do not migrate 1,000 km in a single day.
L44: Delete duplicate “around”
L50-51: “Large flocks of starling(s) are known to destroy fruits in vineyards and orchards and depredate cattle feed at dairy farms.”
L59-64: Reconcile these numbers for the readers. An average of 54,000 birds are removed/year between 2003-2013 yet in 2013 the population in December was 23,592?
L118: Insert “is” after Central Okanagan; insert “of” after up.
L310: spell out kilometer.
L332: delete “and”.
L335: “…that are 100’s of kilometers apart geographically…”
L364-365: “However, some of the problem birds could not be assigned to a source population and were designated as…” “It might be due to a…”
L369: “Third, it is likely…”
L371: “The use of trace elements in bone tissue in identifying the source population of starling(s)…”

·

Basic reporting

Line 84-85: Statement of the null hypothesis. Rather, state the hypothesis.

Figure titles: there are issues with all of them. They need to be stand alone.
For example, if writing 'juveniles' the reader need to know what species. Write 'juvenile starlings'. Same with 'adults'

'locations' = 'locations in the Okanagan Valley, British Columbia'

Do this for all the figure titles.

Figure 3 - '...different the clusters are'. Are what? Are different in relation to each other? the reference samples?

Figure 5 - this is not a heat map.

Experimental design

Yes

Validity of the findings

Yes

Additional comments

Line 63: Bird Studies Canada is now Birds Canada. Also, this reference is not appropriately structured. Did you access this information from the web? It does not appear in the literature cited.

Line 109: do not end a sentence with ect.

Line 395: delete 'like the Okanagan Valley'

---

## Round 0.3 · accepted · Accept

Thank you for a good faith effort to revise. You have met the requirements of the reviewer and I am accepting the paper. Well done.